# Estimation of the Number of Active Sweat Glands Through Discrete Sweat Sensing

**DOI:** 10.3390/s24227187

**Published:** 2024-11-09

**Authors:** Jelte R. Haakma, Elisabetta Peri, Simona Turco, Eduard Pelssers, Jaap M. J. den Toonder, Massimo Mischi

**Affiliations:** 1Department of Electrical Engineering, Eindhoven University of Technology, 5612 AZ Eindhoven, The Netherlands; j.r.haakma@tue.nl (J.R.H.); m.mischi@tue.nl (M.M.); 2Mechanical Engineering, Eindhoven University of Technology, 5612 AZ Eindhoven, The Netherlands; e.g.m.pelssers@tue.nl (E.P.); j.m.j.d.toonder@tue.nl (J.M.J.d.T.); 3Institute for Complex Molecular Systems (ICMS), Eindhoven University of Technology, 5612 AZ Eindhoven, The Netherlands

**Keywords:** health monitoring, sweat, discrete sweat sensing device

## Abstract

Sweat is a biomarker-rich fluid with potential for continuous patient monitoring via wearable devices. However, biomarker concentrations vary with the sweat rate per gland, posing a challenge for sweat sensing. To address this, we propose an algorithm to compute both the number of active sweat glands and their individual sweat rates. We developed models of sweat glands and a discrete sweat-sensing device to sense sweat volume. Our algorithm estimates the number of active glands by decomposing the signal into patterns generated by the individual sweat glands, allowing for the calculation of individual sweat rates. We assessed the algorithm’s accuracy using synthetic datasets for varying physiological parameters (sweat rate and number of active sweat glands) and device layouts. The results show that device layout significantly affects accuracy, with error rates below 0.2% for low and medium sweat rates (below 0.2 nL min^−1^ per gland). However, the method is not suitable for high sweat rates. The suitable sweat rate range can be adapted to different needs through the choice of device. Based on our findings, we provide recommendations for optimal device layouts to improve accuracy in estimating active sweat glands. This is the first study to focus on estimating the sweat rate per gland, which essential for accurate biomarker concentration estimation and advancing sweat sensing towards clinical applications.

## 1. Introduction

Sweat is a relatively unexplored biofluid containing information that can provide broad insights into the metabolic activity of the human body [1,2]. Semicontinuous sweat monitoring can be based on noninvasive sensing technology, representing an attractive candidate for the early detection of patient deterioration. A prompt response to patient deterioration is key to improving patient outcomes and reducing mortality rate [3], as it avoids an increase in morbidity and a (longer) stay in the intensive care unit [3,4].

Sweat is produced by 1.6 to 4.0 million sweat glands (SGs) distributed on the human skin [5]. The sweat gland density varies greatly from person to person [6] and depending on body location, from 10 glands cm^−2^ on the back to 650 glands cm^−2^ on the fingertips, soles, and forehead [5,6]. SGs have a tubular shape, with an opening at the skin to allow the excretion of sweat. The width of the excreting tube is about 40 μm and its length is 2–4 mm [7]. Although current knowledge about the physiological mechanisms of sweat production is limited, there is a consensus that sweat glands are activated by both central and local control mechanisms [6]. The central nervous system initiates synchronized pulsatile sweat production [6,8] with a pattern composed of an active period (*AP*) of about 30 s followed by a resting period of 2–4 min [9]. This pattern additionally influences the number of active SGs [8]. External factors such as heat acclimatization and blood alcohol level also control the amount of sweat that is secreted per active gland [6,10]. The combination of central and external control mechanisms determines the sweat rate on a per-gland basis. The average per-gland sweat rate produced over one minute is widely variable, on the order of 0.02–20 nL/min/gland [11,12]. In a sedentary state, as expected for hospitalized patients, the range can be reduced to about 0.02–0.2 nL/min/gland [11,12]. The overall sweat rate collected on the skin also depends on the number of active SGs ($N_{SG}$). Typically, $N_{SG}$ is determined using the average sweat gland density, which is inherently inaccurate because it differs from person to person and between body locations [7]. Furthermore, it varies over time, as not all sweat glands are active at the same time during an active period [8].

Accurate estimation of $N_{SG}$ is needed in order to compute the average sweat rate *per gland* from the total sweat rate. Indeed, the average sweat rate per gland influences the concentration of several biomarkers, and must be taken into account when interpreting the relationship between biomarkers in blood and sweat [1,6,13]. An example illustrating the relevance of the average sweat rate per gland is lactate, a biomarker that is widely monitored in hospital intensive care units to assess patient deterioration [14]. A portion of the blood lactate is expected to be transported to sweat; however, previous studies have reported very poor blood-to-sweat correlation [6,15]. Lactate is also produced at a high sweat rate per gland by the anaerobic metabolism of the sweat glands themselves [6,16,17]. The contribution of lactate from sweat glands can create a confounding effect that might be responsible for the poor correlation between sweat and blood lactate concentrations reported in multiple previous studies [15].

In recent years, there has been increasing interest in the development of novel sweat-sensing strategies for chemical biomarker monitoring [18]. One recent method proposed the use of microfluidics to realize discretized sweat sensing. The main idea is that sweat is collected in droplets instead of as the continuous flow typically used in traditional devices [18,19]. The main advantage of discrete collection and transport is that it enables semi-continuous monitoring of very small droplet volumes (on the order of nanoliters) and eliminates the inherent diffusion of target molecules between droplets.

This paper proposes a theoretical framework and signal analysis strategy to estimate the number of active sweat glands from the signal acquired by a discretized sweat-sensing device. To this end, the pulsatile sweat generation process and a discretized sweat-sensing device with different designs are both modeled. The device model generally includes sweat collection sites, a sweat droplet transport system, and an ideal droplet volume sensor. Figure 1 provides schematic representations of possible implementations of a discretized sweat-sensing device.

A dedicated signal analysis algorithm is proposed to estimate the number of active sweat glands $N_{SG}$. Its performance is assessed using a synthetic dataset generated through the proposed models. The algorithm estimates the number of active sweat glands $\hat{N_{SG}}$ from synthetic data, with the result then compared to the ground truth $N_{SG}$ under different device designs and sweat rates.

## 2. Materials and Methods

The strategy proposed in this paper is summarized in Figure 2. In this section, we first provide a description of the sweat generation and sweat sensing device models, including a mathematical description of the resulting synthetic signals. Further details on the model design can be found in [20]. Second, we describe the signal analysis algorithm for estimating the number of active sweat glands $\hat{N_{SG}}$, followed by a description of the sources of estimation error. Finally, we report the strategy for assessing the algorithm’s performance on the generated synthetic dataset under different working conditions, i.e., differences in the number of active sweat glands, sweat rate per gland, and device layout. Overall, the proposed strategy is based on a number of assumptions. Within such a simplified framework, we can then compute an estimate of the best achievable algorithm performance so as to provide design recommendations for the development of a discretized sweat-sensing device.

### 2.1. Sweat Generation Model

The sweat generation model simulates the volume of sweat produced by a group of sweat glands. A few assumptions are used based on priors from previous in vivo studies on sweat gland activity [8,9]. Active sweat glands are modeled as randomly distributed over the skin surface [11,12] with an average density of 10 gland/cm^2^, as estimated for the forearm [11]. The model reproduces the pulsatile activity of all sweat glands according to a pattern of 0.5 min of active sweat production ($t_{AP}$) followed by about 2.5 min of resting period $t_{RP}$ [9]. The sweat rate of each sweat gland ($SR_{g}$) is assumed to be constant over one active period of 0.5 min [9] and the same among different sweat glands. Different physiological $SR_{g}$ were simulated ranging from 0.4 to 1 nL/$t_{AP}$/gland during the active period, corresponding to 0.1–0.3 nL/min/gland. The activity of the different sweat glands is considered to be synchronized [7]. The model parameters are summarized in Table 1.

### 2.2. Sweat Sensing Model

The sweat sensing model considers an ideal sweat sensing device positioned on the skin surface to collect, transport, and sense sweat droplets. A possible physical implementation of such a device is reported in Figure 1. When a gland excretes sweat, a droplet is formed on the skin and is collected at the collection site through holes in the back plate that are in contact with the skin (black circles in the figure). When the droplet volume exceeds a minimum threshold ($V_{min}$) and contacts a transport tile (in grey), it can be transported. The actual transport occurs only when the corresponding tile is active (in red). The activation of tiles is mastered through electrical leads. In this paper, we consider five leads $N_{leads}$ that are sequentially activated such that one tile in every five is active simultaneously. The sequential activation of all tiles then transports the droplet toward the volumetric sensor (in green). An example of the physical implementation of this principle in a microfluidic sweat sensing device has been proposed by Moonen through electrowetting on dielectric [19].

In our sweat-sensing model, we consider the following assumptions. First, the device must have a probability higher than 99% to sample sweat from at least one collection site, and each collection site must have a 95% probability of corresponding to at most one sweat gland. This can be achieved by considering the random and independent spatial distribution of active sweat glands with a Poisson distribution [8]. As such, a Poisson distribution can be used to estimate the probability of the number of active sweat glands *n* in a collection site [11]:(1)$$P_{n}=\frac{e^{-\lambda }\lambda^{n}}{n!}$$
with λ corresponding to the average number of sweat glands in the collection area estimated as the product between the sweat gland density (ten sweat glands per cm^2^ on the forearm) and the collection area. Further details can be found in a previous work from our group [11]. As an illustrative example, 5000 collection sites with a diameter of 100 μm on the forearm can be used to meet our assumptions. In this case, the overall collection area is 0.5 cm^2^. Second, we limit our investigation to a range of active sweat glands from 1 to 11. This assumption results from computing λ=5 as the product between the sweat gland density (ten sweat glands per cm^2^ on the forearm) and the overall collection area of the device (0.5 cm^2^). This leads to a probability of having more than 11 sweat glands in the collection area that is smaller than 0.01%. Third, an ideal volumetric sensor is modeled. No other chemical sensors are considered (e.g., lactate or glucose sensors), as this paper is focused on estimating the per-gland sweat rate only.

The model is further characterized by a set of parameters, which are summarized in Table 1. Among these, a pivotal role is played by the minimum droplet volume that can be transported through the activation of subsequent tiles ($V_{min}$). $V_{min}$ depends on the physical dimensions of the tiles in the device, and determines the number of droplets $N_{d}$ collected during the active period *AP* of a sweat gland given its per-gland sweat rate $SR_{g}$ in $\mathrm{nL}/t_{AP}$, as follows:(2)$$N_{d}=\left\lfloor \frac{SR_{g}\cdot t_{AP}+V_{res}}{V_{min}}\right\rfloor$$
where $t_{AP}$ accounts for the duration of the active period, $V_{res}$ represents possible residual sweat volume collected during the previous sweat gland activation, and $V_{min}$ is the minimum droplet volume that can be transported by the device. In this paper, we express the per-gland sweat rate $SR_{g}$ in terms of the number of droplets $N_{d}$ produced during one active period by an individual gland in order to generalize the results for different design choices in terms of $V_{min}$. Given $V_{min}=0.2\mathrm{nL}$ and $SR_{g}=0.4-1\mathrm{nL}/t_{AP}$, $N_{d}$ ranges from 2 to 5 droplets per active period per gland. In general, $SR_{g}$ is translated into a discrete number of droplets, which can vary during one active period depending on the possible residual sweat volume produced during previous active periods ($V_{res}$). For instance, if $SR_{g}=0.7\mathrm{nL}/t_{AP}$, then three or four droplets will be produced with equal probability. In this paper, this sweat rate is reported as $N_{d}=3.5\;droplets/t_{AP}$. In our simulations, we considered seven illustrative sweat rates within the range from the literature, i.e., $SR_{g}=0.4,0.5,0.6,0.7,0.8,0.9,1\mathrm{nL}/t_{AP}/gland$, corresponding to $N_{d}=2,2.5,3,3.5,4,4.5,5$ $droplets/t_{AP}/gland$, respectively. Other relevant model parameters are as follows: (i) the time between two subsequent activations of the same tile, identified as the cycle time ($t_{cycle}$); (ii) the distance a droplet covers from the collection site to the sensor, indicated as the travel distance ($d_{travel}$) and expressed as the number of tiles crossed by the droplet from the collection site to the sensor in a discretized device; (iii) the travel time ($t_{travel}$) of a droplet from the collection site to the sensor, provided by the product of $d_{travel}$ and $t_{cycle}$. The overall layout of the sweat-sensing device can be fully characterized based on the travel distance $d_{travel}$ from each collection site to the sensor. In this paper, we simulate layouts characterized by different uniform distributions of $d_{travel}$. This approach is illustrated in Figure 3a for the three examples of physical devices, represented in panels Figure 3b–d. The depicted devices are simplified for descriptive purposes, with only eight collection sites represented.

In Figure 3b, a simplified device characterized by one single possible $d_{travel}$ repeated eight times is represented, as all the collection sites (black circles) are at the same distance to the sensor. This corresponds to the dark grey distribution in Figure 3a. In Figure 3c, four different $d_{travel}$ values are possible, each repeated two times in the device. In Figure 3d, all eight collection sites correspond to different $d_{travel}$ values, as they are all located at different distances to the sensor. This is represented as light grey in Figure 3a.

As mentioned above, a sweat sensing device is expected to have around 5000 collection sites with a diameter of 100 μm, corresponding to a maximum of one sweat gland per collection site on the forearm. Each of these collection sites corresponds to a $d_{travel}$ value. In our simulations, we chose a number of collection sites $N_{collection}$ equal to 5040. This number is characterized by multiple integer divisors, allowing us to simulate seven layouts with the same number of collection sites but different $d_{travel}$ uniform distributions, as represented in Figure 4a. Furthermore, in order to take into account the role played by the number of collection sites, we also simulated two layouts (I and H) with double the number of collection sites. The distributions are reported in Figure 4b, which are comparable to layouts C and E in terms of the ratio between the $d_{travel}$ and corresponding count values.

All of the described model parameters along with their operational ranges are summarized in Table 1.

### 2.3. Formal Description of the Model Output

In this section, we provide a description of the simulated signals generated by the model. Figure 5 shows how the droplet formation evolves over time at the collection site for two illustrative sweat glands (SG1 in panel a and SG2 in panel b). Each *i*-th gland produces sweat volume $V_{sg_{i}}$ at a per-gland sweat rate $SR_{g}$ during the active period, which is summed with the possible residual volume $V_{res}$ present from the previous active period. When the droplet volume exceeds $V_{min}$, it is available for transportation. The actual droplet transport occurs when the corresponding tile (the grey vertical lines) is activated. The volume sensed by the volumetric sensor is reported in panel c. The difference in the timing of the signals from SG1 and SG2 is due to the difference in the travel time $\Delta t_{travel}$ from the two corresponding collection sites. $\Delta t_{travel}$ is estimated as the time difference between each droplet and the first droplet that arrived at the sensor during the active period. The difference in the timing of two consecutive droplets from the same sweat gland ($\Delta t_{droplet}$) depends on the sweat rate $SR_{g}$.

In general, the sensor signal V(t) corresponds to the overall volumes that originate from multiple sweat glands; V(t) can be modeled as a train of weighted delta pulses, i.e., the summation of the volume signal from each *j*-th droplet coming from the *i*-th sweat glands $V_{sg_{i,j}}$, shifted by the characteristic travel time to the collection site of the *i*-th sweat gland $\Delta t_{travel,i}$ and the timing of the *j*-th droplet from each sweat gland $j\cdot \Delta t_{droplet,i}$, which is provided as follows:(3)$$V\left(t\right)=\sum_{i=1}^{N_{SG}}\sum_{j=1}^{N_{d}}V_{sg_{i,j}}\delta \left(t-\Delta t_{travel,i}-j\cdot \Delta t_{droplet,i}\right)$$
where δ is a unit pulse function that is equal to 1 if its argument is 0 and is 0 otherwise, $N_{SG}$ is the total number of active sweat glands, and $N_{d}$ is the total number of droplets associated with a sweat gland. Clearly, according to Equation (Equation 3), all simulated active sweat glands contribute to the sensor signal. A droplet from an individual sweat gland $V_{sg_{i,j}}$ can merge while being transported to the sensor. In the illustrative example shown in Figure 6, three sweat glands with synchronous activity and the same per-gland sweat rate are simulated. Because two sweat glands (SG1 and SG2) are collected on collection sites with the same $t_{travel}$, their corresponding volumes $V_{sg_{1,1}}$ and $V_{sg_{2,1}}$ are merged to create a droplet with volume $V_{sg_{1,1}}+V_{sg_{2,1}}$. The third sweat gland is collected from a collection site with a longer $t_{travel}$, resulting in a delayed volume at the sensor. This delay corresponds to the difference in travel time between the collection site of SG3 ($t_{travel,3}$) with respect to the collection sites corresponding to the sweat glands that produced the first droplets to arrive at the sensor ($t_{travel,1}=t_{travel,2}$).

The volume of each *j*-th droplet from the *i*-th sweat gland is mainly determined by the per-gland sweat rate $SR_{g}$, cycle time $t_{cycle}$, and minimum droplet volume $V_{min}$, with its range determined as follows:(4)$$V_{min}\leq V_{sg_{i,j}}<V_{min}+SR_{g}\cdot t_{cycle}.$$

The exact volume of the first droplet of each active period ($V_{sg_{i},1}$) may differ with respect to other droplets from the same sweat gland depending on the possible residual sweat volume $V_{res}$ from the previous active period, as shown in Figure 5.

### 2.4. Estimating the Number of Active Sweat Glands

As the sweat rate per gland is unknown for human subjects, estimating the number of active sweat glands solely from the total sweat rate observed by the sensor is not possible. In this paper, we estimate the number of active sweat glands by analyzing the patterns in the volume signal V(t) corresponding to the physiological pulsatile sweat gland behavior. This pattern depends on regular intervals between non-merged droplets, i.e., the droplets produced by each individual sweat gland $V_{sg_{i}}$. The proposed algorithm for deriving the number of active sweat glands consists of three steps: segmentation, quantization, and decomposition.

#### 2.4.1. Segmentation

Segmentation is used to segment the signal into windows, each containing information on an individual active period. This step allows for estimation of the number of active sweat glands and corresponding sweat rate per gland separately for each active period. Segmentation is performed considering that the maximum duration of a window is equal to the sum of the duration of the active period (30 s) and the difference between the maximum and minimum travel time that characterizes the simulated device layout.

#### 2.4.2. Quantization

Quantization is used to determine the volume of non-merged droplets from one sweat gland before they enter the transport system.

To this end, we consider every droplet at the sensor to be composed of an integer number of non-merged droplets with approximately the same volume, as modeled in this paper. We compute the ratio *r* between the measured droplet volume V(t) and a set of possible non-merged droplet volumes distributed within the range defined in Equation (Equation 4). The droplet volume $\hat{V_{sg}}$ which minimizes the sum of the squared differences between the computed *r* and its closest integer is identified. This value is then chosen to quantize the volume, as follows:(5)$$V_{q}\left(t\right)=\frac{V(t)}{\hat{V_{sg}}}.$$

#### 2.4.3. Signal Decomposition

Signal decomposition is aimed at estimating the number of active sweat glands $N_{SG}$ by separating their individual patterns. In this step, each window defined in the segmentation step is analyzed separately; $N_{SG}$ is determined by fitting the quantized signal $V_{q}\left(t\right)$ using the model in Equation (Equation 3), as follows:(6)$$\min_{N_{SG},N_{d}}{\left|\left|V_{q}\left(t\right)-\sum_{i=1}^{N_{SG}}\sum_{j=1}^{N_{d}}\delta \left(t-\Delta t_{travel,i}-j\cdot \Delta t_{droplet,i}\right)\right|\right|}^{2}$$
where $N_{SG}$ is the total number of active sweat glands, $N_{d}$ is the total number of droplets associated with a sweat gland, $\Delta t_{travel,i}$ is the difference between the travel time of the *j*-th droplet and the first droplet arriving at the sensor during the active period, and $\Delta t_{droplet,i}$ is the difference in time between the production of two consecutive droplets from the *i*-th sweat gland. An exhaustive search of the whole parameter space is performed. To limit the search space, four sets of parameter boundaries are used, considering the following constraints described: (1) $N_{SG}$ ranges from 1 to 11 sweat glands. (2) The range of $\Delta t_{travel,i}$ depends on the device design. With reference to Figure 4, $\Delta t_{travel,i}$ ranges between 0 s for layout G, where all of the collection sites have the same travel time $t_{travel}$, to 198 s for layout A and $t_{cycle}=0.75\;$s, characterized by 1260 different $t_{travel}$ values. (3) The ranges of $N_{d}$ and $\Delta t_{droplet,i}$ were jointly chosen such that (i) a minimum of two droplets per active period (30 s) per gland are present to create a sweat-gland specific pattern, and (ii) a maximum of one droplet for the tile activation of each collection site ($t_{cycle}$ = 0.25–0.75 s) is created to maintain the droplet volume within the ranges defined in Equation (Equation 4). This results in $N_{d}$ ranging from 2 to 120 and $\Delta t_{droplet,i}$ ranging from 0.25 s to 15 s. To reduce the computational time, the values spanned by $\Delta t_{droplet,i}$ are restricted to candidate intervals present in the signal. An illustrative visualization of the principle used is reported in Figure 7, where all of the time differences between a segment’s first and subsequent peaks up to and including half of the active period were chosen as candidate $\Delta t_{droplet,i}$ (in black). Intervals not satisfying the parameter boundaries described above were not considered, e.g., $\Delta t_{droplet,i}=21$ s (in grey), which did not satisfy the requirement of having a minimum of two droplets during one active period of 30 s.

The number of sweat glands $\hat{N_{SG}}$ that produces a match between $V_{q}\left(t\right)$ and its model is considered a possible solution. In some cases, more solutions were possible given a certain signal $V_{q}\left(t\right)$. In such cases, $V_{q}\left(t\right)$ was considered to be an ambiguous signal.

### 2.5. Failures and Ambiguous Signals

There were a number of cases in which the described approach was unable to estimate a unique number of active sweat glands $N_{SG}$ because the same segment $V_{q}\left(t\right)$ could be obtained from different numbers of active sweat glands. When the algorithm failed to return a solution or when none of the returned solutions was correct, the segment is referred to as a “failure signal”. When multiple solutions were returned and only one was correct, the signal segment is referred to as an “ambiguous signal”.

Several factors influence the frequency at which ambiguous signals occur, including the layout and cycle time, which influences the estimation error. In this paper, we classify ambiguities into type (1) ambiguous quantization, type (2) ambiguous $N_{d}$, and type (3) ambiguous $\Delta t_{droplet}$, as summarized in Figure 8. It is noteworthy that a single signal can belong to more than one of these groups.

#### 2.5.1. Ambiguous Quantization

Ambiguous quantization occurs when the volume of the non-merged droplet cannot be uniquely determined in the quantization step due to multiple possible options within the acceptable volume range defined in Equation (Equation 4). In the illustrative example reported in Figure 8a, if the sensor is measuring a droplet of volume *V*, then the algorithm cannot uniquely determine whether this volume has been obtained by merging two non-merged droplets each with volume V/2, or three non-merged droplets each with volume V/3. This issue occurs when both V/2 and V/3 are within the acceptable volume range defined in Equation (Equation 4). Ambiguous quantization depends on the probability of merging in a particular device design, and becomes more critical at higher sweat rates (four droplets/*AP* or more), as droplet merging occurs more often under these conditions.

#### 2.5.2. Ambiguous $N_{d}$

Ambiguous $N_{d}$ occurs when at least two of the decomposed patterns have the same interval $\Delta t_{droplet}$. The example in Figure 8b shows that the same signal can be simulated considering two sweat glands each producing three droplets, or three sweat glands each producing two droplets. The chance of ambiguous $N_{d}$ is expected to increase as the number of active sweat glands increases.

#### 2.5.3. Ambiguous $\Delta t_{droplet}$

Ambiguous $\Delta t_{droplet}$ occurs when the decomposition of a signal leads to multiple results that differ in the number of droplets per gland $N_{d}$ as well as in the interval between two droplets from the same gland $\Delta t_{droplet}$. In the illustrative example reported in Figure 8c, the signal could be obtained by simulating one individual sweat gland with a given sweat rate (on the right) or two sweat glands with a halved sweat rate. Thus, an increase in the sweat rate per gland is expected to increase the frequency at which ambiguous $\Delta t_{droplet}$ signals occur. At the same time, the probability of a pattern of evenly spaced droplets being created is smaller when multiple sweat glands are active. Therefore, the frequency of ambiguous $\Delta t_{droplet}$ is expected to decrease as the number of active sweat glands increases.

### 2.6. Datasets and Performance Metrics

To assess the performance of the proposed algorithm under different working conditions, several scenarios accounting for different combinations of the model parameters in Table 1 were simulated. The performance of the algorithm in each scenario was tested using 2000 simulated signals, with each signal corresponding to the sweat volume produced by a random number of sweat glands between 1 and 11 randomly assigned to a collection site. The duration of each simulated signal was equal to 15 min, which corresponds to five segments each representing a full sweat gland activity cycle of 3 min.

A total of $10^{4}$ segments representing sweat gland activity cycles were simulated and used to estimate the error rate. For each of these segments, the signal was decomposed to derive the number of sweat glands that contributed to the signal. When a segment could be decomposed in multiple ways, the algorithm resulted in a list of possible $\hat{N_{SG}}$.

The error rate was calculated as the mean of the percentage error per segment, as follows:(7)$$Error\;rate=\sum_{s=1}^{S}\frac{E_{s}}{S}$$
with *S* as the total number of segments and $E_{s}$ calculated as
(8)$$E_{s}=\frac{N_{incorrect}}{N_{total}}\cdot 100,$$
where $N_{incorrect}$ and $N_{total}$ are the number of incorrect estimations and total number of estimations, respectively, for one segment. Note that multiple estimations can occur in case of ambiguous signals; thus, $E_{s}$ represents the probability of the estimated $\hat{N_{SG}}$ being incorrect for that segment, with $E_{s}=0\%$ meaning that the number of glands was correctly and uniquely derived from the segment and $E_{s}=100\%$ meaning that no estimations were correct. The error rate is reported along with its 95% confidence interval, which was computed by means of a two-sided 95% bias-corrected and accelerated bootstrap with $10^{5}$ bootstrap replicators [21].

## 3. Results

### 3.1. Effect of Physiological Parameters $SR_{g}$ and $N_{SG}$

The error rate in estimating the number of active sweat glands as a function of the real $N_{SG}$ for low ($N_{d}=2\;droplets/t_{AP}$, $SR_{g}=0.4\mathrm{nL}/t_{AP}$), medium ($N_{d}=3.5\;droplets/t_{AP}$, $SR_{g}=0.2\mathrm{nL}/t_{AP}$), and high ($N_{d}=5\;droplets/t_{AP}$, $SR_{g}=0.3\mathrm{nL}/t_{AP}$) sweat rates is shown in Figure 9. The results are reported for $T_{cycle}=0.5$ s and layouts A–I, with the exception of layout G. Simulations with layout G obtained error rate that were always either one or zero, and as such these results are excluded from the figures for improved readability. The error rate increases as the sweat rate increases, and is always below 7% for low and medium sweat rates (panels a and b). Overall, the error rate trend for increasing number of active sweat glands shows a peak for $N_{SG}=2$ or 3 followed by decrease, with the exception of layout A. In panel c, the error rate at high sweat rates shows values consistently higher than 50% for all simulated active sweat glands as well as for all layouts.

The effect of sweat rate on the error rate for illustrative device layout D is represented in Figure 10. Here, as the sweat rate increases from $N_{d}=3.5\;droplets/t_{AP}/gland$ to $N_{d}=4\;droplets/t_{AP}/gland$, the error rate increases from less than 10% to more than 50%. At $N_{d}=5\;droplets/t_{AP}/gland$, the error rate increases as $N_{SG}$ increases. Similar behavior is present for all layouts.

### 3.2. Effect of Device Design

#### 3.2.1. Device Layout

The device layout influences the results, as shown in Figure 9. Layouts D, E, and F show lower average error (below 0.2%) compared to the other layouts at low and medium sweat rates ($N_{d}<4\;droplets/t_{AP}/gland$), while the performance of all layouts is comparable at high sweat rates (above 50%). Device layouts A and G show significantly different patterns than the other device layouts. Layout A is already characterized by an increase in the error rate as the number of active sweat gland increases at low and medium sweat rates. Layout G obtains an error rate above 50% for all sweat rates when more than two sweat glands are active. Layouts A, B, C, and H show local worsening in performance when two sweat glands are active. The number of collection sites does not significantly affect the average error rate; indeed, the devices characterized by 5040 collection sites (layouts B–F) and 10,080 collection sites (layouts H and I) obtain average error rates below 1% at low and medium sweat rates and between 50–60% at high sweat rates.

#### 3.2.2. Cycle Time $t_{cycle}$

The cycle time of the device also significantly affects the algorithm’s performance. The results shown in Figure 11 highlight that the average error rate with 2–11 active sweat glands increases as the time between two subsequent activations of the same tile increases, up to the highest average error rate with $t_{cycle}=0.75\;$s of 10% at low to medium sweat rates ($N<4\;droplets/t_{AP}/gland$). At higher sweat rates, the error rate is above 50% independent of $t_{cycle}$.

Furthermore, the error rates of different layouts are affected differently by changes in the cycle time. Layouts D, E, and F are less affected by changes in $t_{cycle}$. Figure 12 shows the combined effect of the different layouts and different $t_{cycle}$ for the illustrative case of a sweat rate of $3.5\;droplets/t_{AP}/gland$.

### 3.3. Failure and Ambiguous Signals

The error rate is determined by failures and ambiguous signals, i.e., those for which our algorithm is not able to return a (unique) correct estimation of the number of active sweat glands.

Failure signals, i.e., signals for which no result is returned by the algorithm, occurred only when simulating sweat production with layouts A and G. For the dataset simulated with layout A in combination with $t_{cycle}=0.75\;$s, the failure rate was caused by a failure in the segmentation step, and was so high that this scenario was not explored further. In addition, the simulations with layout G had 100% failure signals when four or more active sweat glands were considered. The other scenarios had no failure signals.

Figure 13 shows the frequency of occurrence of ambiguous signals due to ambiguous quantization, $N_{d}$, and $\Delta t_{droplet}$ in simulation with layout D and cycle times $t_{cycle}=0.5\;$s as a function of $N_{SG}$ at different sweat rates: low ($N_{d}=2\;droplets/t_{AP}$, $SR_{g}=0.4\mathrm{nL}/t_{AP}$), medium ($N_{d}=3.5\;droplets/t_{AP}$, $SR_{g}=0.2\mathrm{nL}/t_{AP}$), and high ($N_{d}=5\;droplets/t_{AP}$, $SR_{g}=0.3\mathrm{nL}/t_{AP}$). At low and medium sweat rates (panels (a) and (b)), the dominant source of error was due to an ambiguous quantization (type 1 ambiguity), which entails ambiguous determination of $N_{d}$ (type 2 ambiguity). At high sweat rates (panel (c)), the algorithm always provided more solutions due to ambiguous $\Delta t_{droplet}$ estimation (type 3 ambiguity) for all the signals. Furthermore, the frequency of error due to ambiguous $N_{d}$ (type 2) increases as $N_{SG}$ increases.

The complete results on the error-rate are reported in Appendix A.

## 4. Discussion

In this paper, we have proposed a first approach for estimating the number of active glands contributing to the production of sweat. We tested our algorithm on a synthetic dataset simulating sweat production, collection, and sensing through a discretized sweat sensing device. The error rate in estimating the number of active sweat glands was determined considering that some signals may be ambiguous, i.e., they could be obtained from a number of different active sweat glands.

The frequency at which multiple $\hat{N_{SG}}$ are estimated is influenced by four factors, each of which can be either influenced or chosen when designing a physical sweat-sensing device: (1) the number of droplets per active period per gland ($N_{d}$) collected through the device; (2) the number of active sweat glands ($N_{SG}$) sampled through the device; (3) the layout used in the design of the sweat sensing device; and (4) the cycle time duration ($t_{cycle}$). These factors are not independent.

### 4.1. Effect of Physiological Parameters

#### 4.1.1. Per-Gland Sweat Rate

The discretized sweat sensing principle allows for translation of the per-gland sweat rate to the number of droplets per gland for the active period $N_{d}$. The sweat rate is the major determinant of our algorithm’s performance. As can be observed in Figure 10, the error rate is acceptable only below $4\;droplets/t_{AP}/gland$; above this per-gland sweat rate, the obtained error rate is higher than 50%.

When four or more droplets are produced by each gland during one active period, the sensor signal presents consistently ambiguous signals of type 3, even when only one sweat gland is active. This can be seen in Figure 13c, where the algorithm cannot uniquely identify $\Delta t_{droplet}$ (type 3 ambiguity) for all of the signals. This is because the same signal can be produced by one sweat gland producing $N_{d}=4\;droplets/t_{AP}/gland$ or by two sweat glands producing $Nd=2\;droplets/t_{AP}/gland$. In this case, the error rate reflects that there is one correct estimation with two possible solutions, creating a lower bound of 50%.

Under such high sweat rates, the algorithm does not perform satisfactorily; however, the number of droplets entering a sweat-sensing device at each sweat rate depends on the minimum droplet volume $V_{min}$ that can be collected and transported toward the sensor. Depending on the operational range of the specific application, a different $V_{min}$ can be chosen by changing the size of the transport tiles. Importantly, the device specifications indicated in the current paper were identified by considering long-term monitoring of hospitalized patients, for whom the target sweat rate per gland is expected to be around 0.2 nL min^−1^ [12]. Under such conditions, $V_{min}=0.2\mathrm{nL}$ would allow us to estimate the number of active sweat glands for a sweat rate range between $SR_{g}$ = 0.1–0.3 nL min^−1^ for each sweat gland. For target applications in different fields characterized by larger sweat rates, e.g., sports science, a larger $V_{min}$ can be chosen to guarantee that less than $4\;droplets/t_{AP}/gland$ are produced and that $N_{SG}$ can be estimated correctly.

#### 4.1.2. Number of Active Sweat Glands $N_{SG}$

Our obtained results demonstrate that the real number of active sweat glands contributing to the signal affects the estimated number of active sweat glands. For certain device designs, e.g., layout A, the increase in the number of active sweat glands leads to a higher error rate, as can be seen in Figure 9 and Figure 12. This is predominantly caused by an increase in the frequency of ambiguous $N_{d}$ signals (type 2). For other scenarios, e.g., layout D, $N_{d}=5\;droplets/t_{AP}/gland$, the frequency of signals with ambiguous $N_{d}$ (type 2) rises significantly as $N_{SG}$ increases. However, in the other scenarios where the travel times are often different for different collection sites, e.g., layout B, the increase in the error rate is negligible.

Different scenarios are characterized by a peak in the error rate when two sweat glands are active. In these cases, the number of ambiguous estimations due to ambiguous $\Delta t_{droplet}$ (type 3) is responsible for this behavior. Type 3 ambiguities depend on the fact that multiple possible intervals between droplets ($\Delta t_{droplets}$) match the layout-specific difference in travel time ($\Delta t_{travel}$). When the device layout does not allow this matching $\Delta t_{travel}$, the error rate at two active sweat glands is not present. For instance, layouts A, B, C, and H are all characterized by collection sites with a large maximum difference in travel time. As such, the error rate is zero when two sweat glands with a low sweat rate per gland are active ($N_{d}=2\;droplets/t_{AP}/gland$), as reported in Figure 9a. The sweat rate increases as $\Delta t_{droplets}$ decreases, resulting in smaller matching $\Delta t_{travel}$. Because more layouts allow this match at a higher sweat rate, the peak at two active sweat glands can be seen for more layouts, as shown in Figure 9b,c. It is noteworthy that $t_{cycle}$ can influence this mechanism due to its impact on the computation of $\Delta t_{travel}$.

### 4.2. Effect of Device Parameters

#### 4.2.1. Influence of the Layout

The layout clearly impacts the error rate, as it influences the chance of ambiguous signals arising. In most layouts, the number of occurrences of each travel time is substantially larger than the number of simulated sweat glands, with layout A being the exception. With only four occurrences of each travel time, the number of occurrences is no longer much larger than the number of simulated sweat glands. This could explain why layout A shows some peculiar behavior in Figure 9.

Layouts D and I are very close in terms of the number of unique travel times/distances (i.e., $d_{travel}$ of 70 vs. 72, as shown in Figure 4). Thus, the chance of a droplet being assigned to a given collection site is very similar for those layouts, as are the travel distance and travel time. This corresponds to a similar chance of ambiguous signal occurrence, and consequently to a comparable error rate (see Figure 9). The same holds for layouts B and H ($d_{travel}$ of 144 and 140).

#### 4.2.2. Influence of Cycle Time

A comparison of the three different graphs in Figure 11 and Figure 12 illustrates how the cycle time influences the error rate. In general, shorter cycle times lead to reduced error rates. This general trend is expected, as a higher cycle time increases the number of cycles per active period, essentially increasing the resolution and consequently reducing the chance of the same specific patterns being formed.

### 4.3. Failure Signals

Failure signals, i.e., signals that do not allow the correct solution to be obtained from among the possible outputs of the proposed algorithm, occur in only two scenarios. Specifically, all of the signals simulated using layout G with more than five active sweat glands consisted exclusively of failure signals. As layout G has only one travel time, all droplets merge in this design. This causes the quantization step of the algorithm to fail to produce an estimate. The second scenario involves layout A and $t_{cycle}=0.75\;$s. Here, the number of failure signals is more than 50% of the considered signals. Layout A has many different travel distances, allowing for large differences in travel times. In this case, droplets from one active period may be interpreted as droplets from the successive active period, causing the segmentation step to fail.

### 4.4. Design Recommendations

Our work shows that three main design parameters must be considered in order to estimate the per-gland sweat rate using a discrete sweat-sensing device. First, tuning $V_{min}$ is key to matching the sweat rate range for the specific application while ensuring that the maximum number of droplets produced by one individual sweat gland does not exceed $4\;droplets/t_{AP}$. Indeed, the proposed algorithm can only be used for $N_{d}<4\;droplets/t_{AP}/gland$. Second, the resolution of the signal is determined by $t_{cycle}$, which should be as low as possible in order to reduce the probability of multiple different combinations of sweat gland activity producing the same pattern. Lastly, designs D, E, and F are characterized by similar travel distances for more collection sites, and seem to obtain a lower average error rate for all cycle time combinations.

### 4.5. Limitations and Future Work

The presented work is the first step towards the development of a strategy for estimating the per-gland sweat rate in combination with the development of discrete sweat sensing. Several aspects can be improved in future work. First, the performance of the proposed algorithm could be improved by selecting the most probable solution from among the provided solutions. Especially at higher sweat rates, where the error rate reaches 50%, great improvements can be made. Second, the model used to generate the signals was mostly deterministic, with the only stochastic component consisting of the distribution of the sweat glands over the collection sites. In the future, a more advanced simulator could be developed that incorporates stochastic descriptors to accurately describe the behavior of sweat glands, including variable sweat rate per gland [22] (both between glands [23] and during the active period), non-synchronized sweat gland activation, and variable duration of the active and rest periods [9]. Third, the modeled device is ideal, i.e., it does not consider sensor noise or possible failure of the collection and transport systems. While incorporating non-ideal sensed volume might affect the quantization step of the algorithm, non-ideal collection and transport of droplets might have consequences on the pattern decomposition. In this paper, we have focused on ideal device characteristics as a first step in investigating the possibility of estimating the number of active sweat glands with discrete sweat sensing in a simplified setting. Future work will build on our current findings to propose strategies that can deal with signals from non-ideal discrete sweat-sensing devices. The approach presented in this paper is based on in silico data only. This step is necessary, as there is no alternative way to directly assess the ground truth in terms of the number of active sweat glands. Future development of discrete sweat-sensing devices such as proposed in [19] would allow for the optimization and validation of our algorithm using real-world data starting from the recently developed artificial skin platform.

## 5. Conclusions

The estimated the number of active sweat glands can be used to determine the per-gland sweat rate, a key metric for clinical interpretation of biomarker concentrations in sweat. This study presents a novel approach for estimating the number of active sweat glands using synthetic signals from a discretized sweat-sensing device. The estimation accuracy is evaluated considering the effect of physiological parameters as well as device design parameters. The proposed method and achieved results provide guidance on the design of digital sweat sensing devices and set the framework for future work incorporating real-world experimental data.

## Figures and Tables

**Figure 1 sensors-24-07187-f001:**
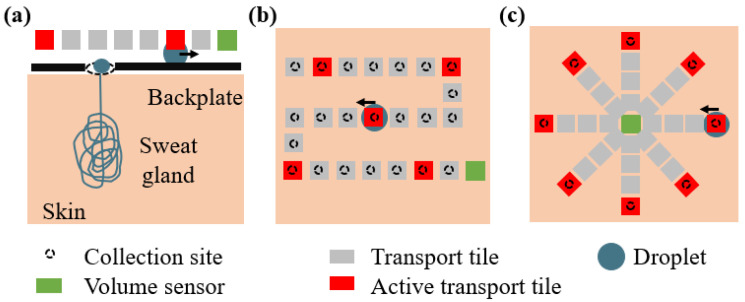
Schematic representation of the proposed discretized sweat sensing device. The collection sites (black circles) allow sweat to enter the device. The formed droplet is transported by transport tiles (in grey) as soon as they are activated (in red). The sequential activation of all tiles transports the droplet toward the sensor (in green). Panel (**a**) shows the cross-section of the discretized sweat-sensing device, while panels (**b**,**c**) show top views of two possible illustrative design layouts. In (**b**), the collection sites are distributed along the transport system such that the droplet at each collection site has a unique distance to the sensor (shown in green). In (**c**), the droplets at each collection site are at an equal distance from the sensor.

**Figure 2 sensors-24-07187-f002:**
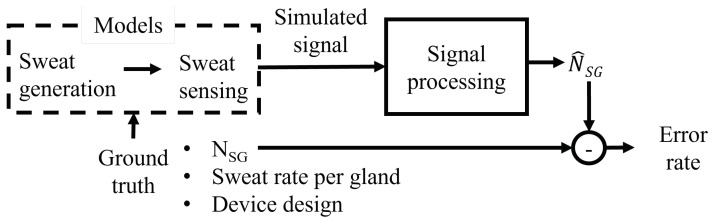
Block diagram of the proposed signal processing chain. The models of sweat generation and sensing are used to produce a synthetic volume signal based on the given number of sweat glands ($N_{SG}$), sweat rate per gland, and device design as inputs. The simulated signal is subsequently fed to the signal processing algorithm, which estimates the number of sweat glands $\hat{N_{SG}}$. The difference between $\hat{N_{SG}}$ and $N_{SG}$ is used to obtain the error rate.

**Figure 3 sensors-24-07187-f003:**
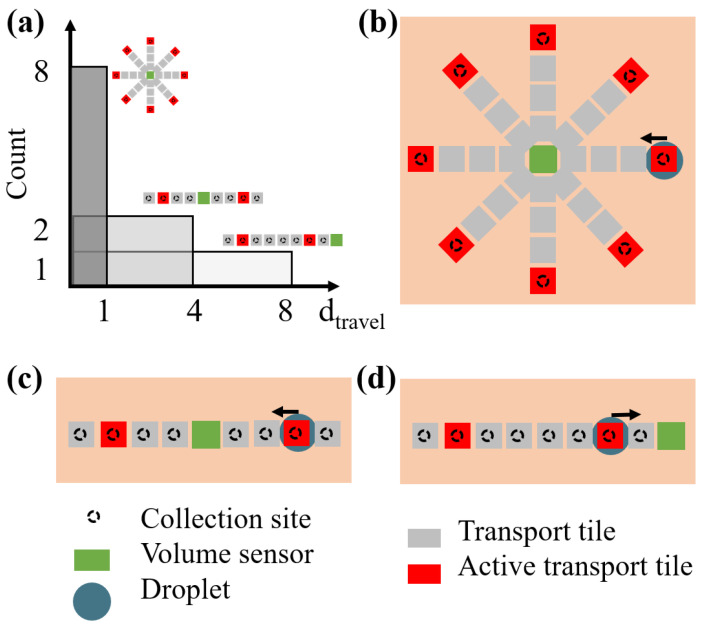
Three illustrative travel distance distributions are represented in panel (**a**), along with three possible simplified design layouts of the physical device (panels (**b**–**d**)). Panel (**b**) illustrates a design layout characterized by all eight collection sites having the same travel distance (dark grey in panel (**a**), panel (**c**)) illustrates a design layout characterized by four possible travel distances, and panel (**d**) illustrates a design layout characterized by all collection sites having different travel distances (light grey in panel (**a**)).

**Figure 4 sensors-24-07187-f004:**
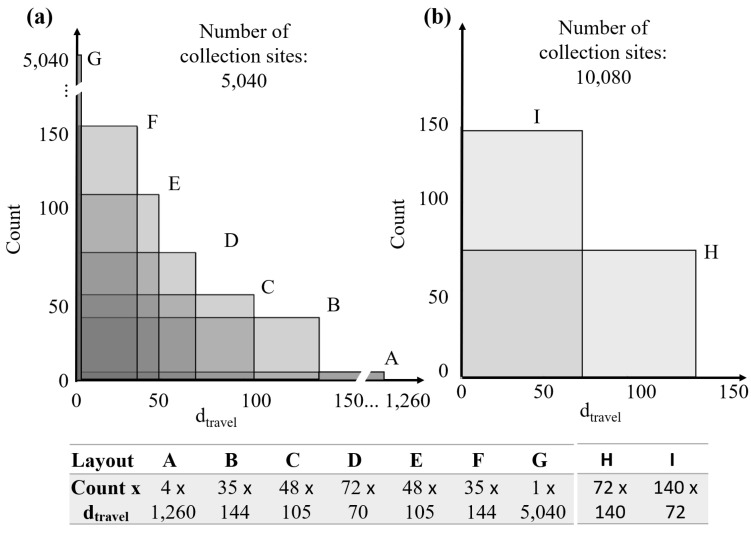
Simulated layouts as characterized by different uniform distributions of the travel distances $d_{travel}$. Panel (**a**) $N_{collection}$ = 5040 collection sites for layouts A–G. Panel (**b**) $N_{collection}$ = 10,080 collection sites for layouts H and I. Simplified illustrative physical representations of layout A, D and G are reported in Figure 3.

**Figure 5 sensors-24-07187-f005:**
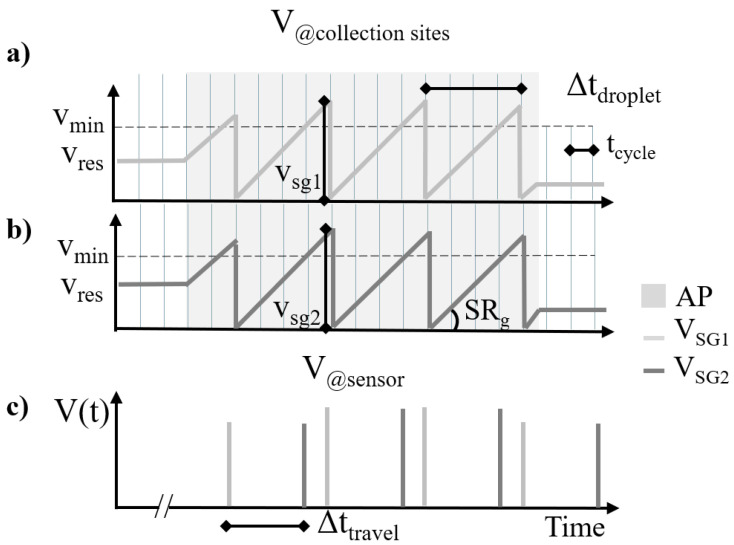
Panels (**a**,**b**) show the sweat volume corresponding to the active period (*AP*, grey area) of sweat glands SG1 and SG2 in two collection sites as a function of time. The slope represents the per-gland sweat rate $SR_{g}$. The grey vertical lines mark the times at which the transport system can transport a droplet from the collection site towards the sensor. This occurs only if the droplet has reached the minimum required volume $V_{min}$, represented as a dotted line. The time interval between tile activations is $t_{cycle}$. Panel (**c**) illustrates the signal sensed by the volumetric sensor. The delay between the signals from the two sweat glands is due to the difference in travel time $\Delta t_{travel}$ between collection sites. $\Delta t_{droplet}$ is the difference in the timing of two consecutive droplets from the same sweat gland.

**Figure 6 sensors-24-07187-f006:**
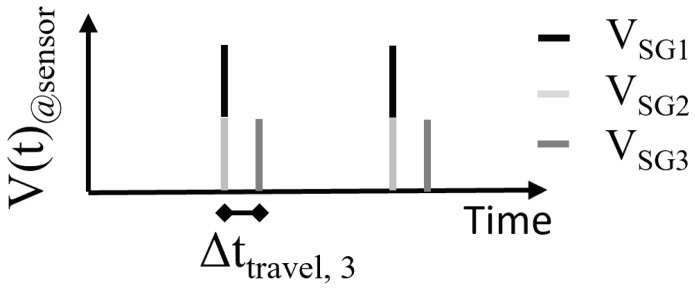
A synthetic signal from three sweat active glands, each with the same sweat rate $SR_{g}$, over one active period. Two of the sweat glands (SG1 and SG2) have the same travel time to the sensor; for this reason, the droplets are merged while being transported toward the sensor. The third sweat gland (SG3) is collected at a site characterized by a longer $t_{travel}$, resulting in $\Delta t_{travel,3}$. For this reason, the non-merged droplet arrives at the sensor later.

**Figure 7 sensors-24-07187-f007:**
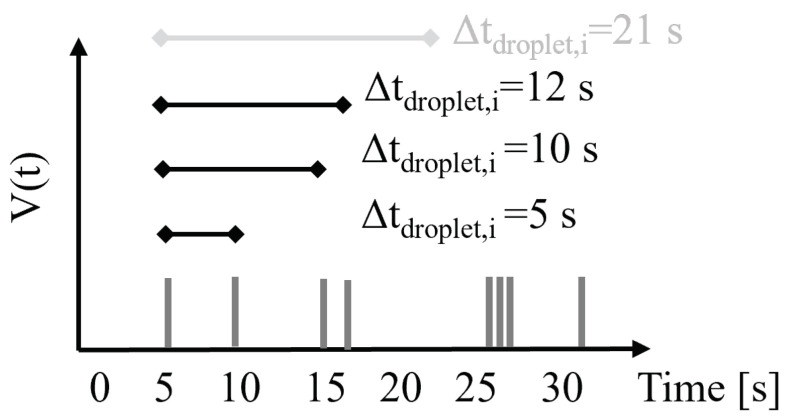
Example of the signal from five active sweat glands over one active period of 30 s. Each peak represents a droplet passing the volumetric sensor. The different candidate intervals $\Delta t_{droplet,i}$ from the *i*-th sweat gland included in the grid search are represented. Candidate $\Delta t_{droplet,i}$, which failed to meet the criteria of maximum interval length and consequently was not used in the analysis, is shown in light grey.

**Figure 8 sensors-24-07187-f008:**
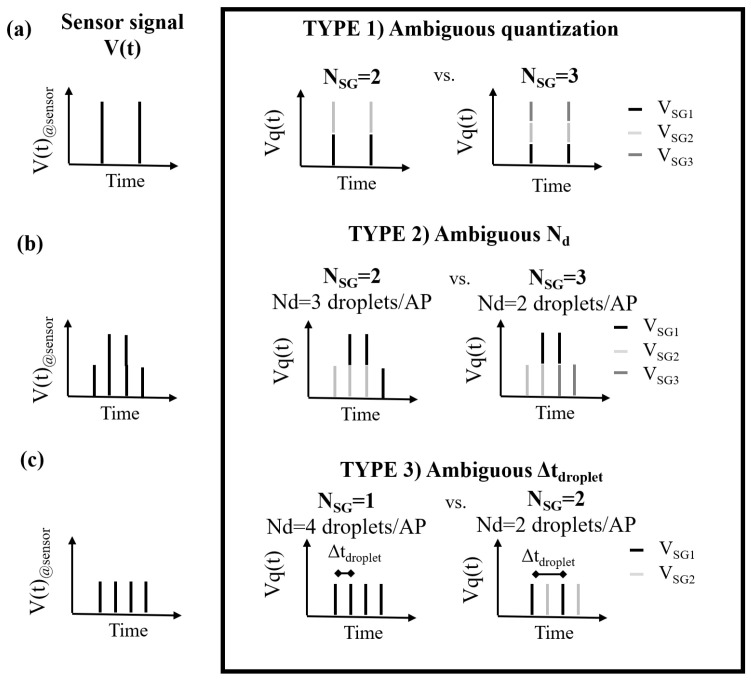
Illustrative ambiguous signals. The same signals V(t) (on the left) can be decomposed into different patterns coming from different numbers of active sweat glands. In panel (**a**), ambiguous quantization results in incorrect estimation of the non-merged droplet volume, and consequently in two possible $N_{SG}$ (right: two active sweat glands; left: three active sweat glands). In panel (**b**), an ambiguous $N_{d}$ signal leads to the choice of two different sweat rates per gland, expressed as the number of droplets per gland $N_{d}$ corresponding to the estimation of different $N_{SG}$ (right: two active sweat glands; left: three active sweat glands). In panel (**c**), an ambiguous $\Delta t_{droplet}$ signal leads to the same signal being decomposed into different sweat rate patterns corresponding to different $\Delta t_{droplet}$, resulting in two possible $N_{SG}$ (right: one active sweat gland; left: two active sweat glands).

**Figure 9 sensors-24-07187-f009:**
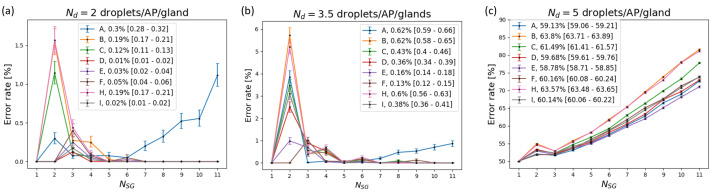
Error rates as a function of the number of active sweat glands at (**a**) low ($2\;droplets/t_{AP}/gland$), (**b**) medium ($3.5\;droplets/t_{AP}/gland$), and (**c**) high ($5\;droplets/t_{AP}/gland$) sweat rates for layouts A–I at a cycle time of 0.5 s.

**Figure 10 sensors-24-07187-f010:**
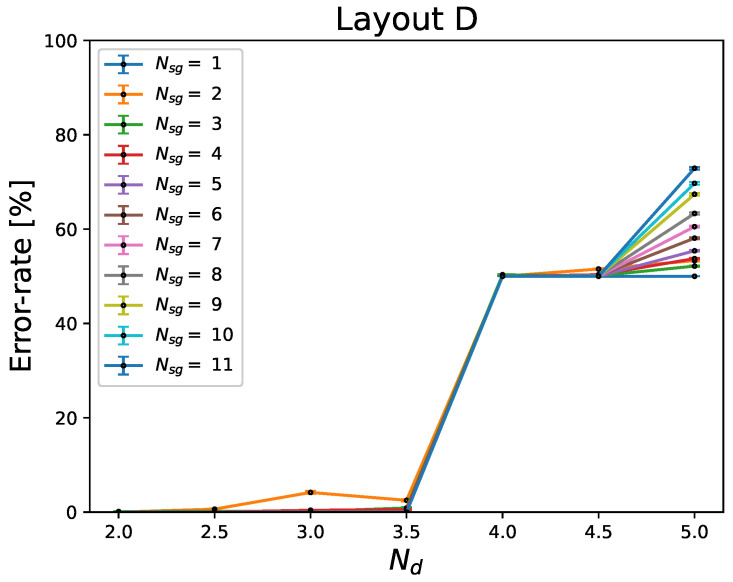
Error rate of layout D as a function of the sweat rate, expressed as the number of droplets per active period. The different colors represent the different numbers of active sweat glands (2–11). $t_{cycle}=0.5\;$s.

**Figure 11 sensors-24-07187-f011:**
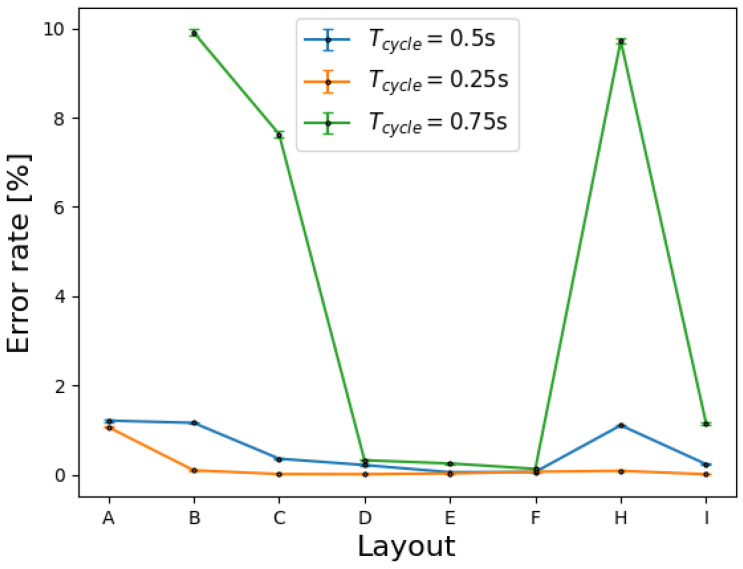
Average error rates with different numbers of active sweat glands (2–11) for different layouts and cycle times: $t_{cycle}=0.25\;$s (orange), $t_{cycle}=0.50\;$s (blue), and $t_{cycle}=0.75\;$s (green) for $N_{d}<4\;droplets/t_{AP}/gland$.

**Figure 12 sensors-24-07187-f012:**
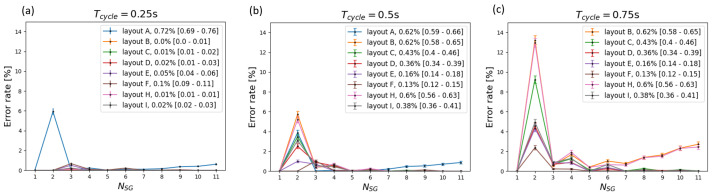
The error rate as a function of the number of active sweat glands for different layouts at $t_{cycle}=0.25\;$s (panel **a**), $t_{cycle}=0.50\;$s (panel **b**), and $t_{cycle}=0.75\;$s (panel **c**) for $N_{d}=3.5\;droplets/t_{AP}/gland$.

**Figure 13 sensors-24-07187-f013:**
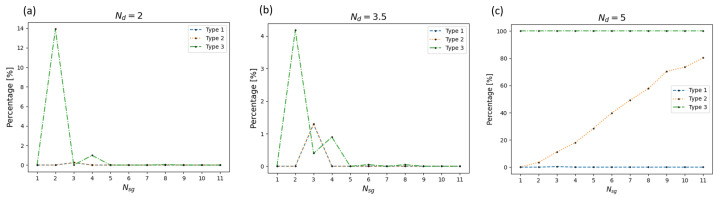
Frequency of occurrence of ambiguous signals due to ambiguous quantization (Type 1—blue), ambiguous $N_{d}$ (Type 2—orange), and ambiguous $\Delta t_{droplet}$ (Type 3—green) in simulations with layout D and cycle time $t_{cycle}=0.5$ s for (**a**) low ($N_{d}=2\;droplets/t_{AP}/gland$, $SR_{g}=0.4\;\mathrm{nL}/t_{AP}/gland$), (**b**) medium ($N_{d}=3.5\;droplets/t_{AP}/gland$, $SR_{g}=0.2\;\mathrm{nL}/t_{AP}/gland$), and (**c**) high ($N_{d}=5\;droplets/t_{AP}/gland$, $SR_{g}=0.3\;\mathrm{nL}/t_{AP}/gland$) sweat rates.

**Table 1 sensors-24-07187-t001:** Model parameters.

Sweat generation model	
$t_{AP}$	30 [s]
$t_{RP}$	150 [s]
$N_{SG}$	1:11
$SR_{g}$	0.4–1 $\left[\mathrm{nL}/t_{AP}/gland\right]$
Sweat sensing model	
$N_{leads}$	5
$V_{min}$	0.2[nL]
$N_{collection}$	5040–10,080
$t_{cycle}$	0.25–0.75 [s]
$d_{travel}$	1–1260 [tiles]
$t_{travel}$	0.25–945 [s]

## Data Availability

The datasets presented in this article are not readily available because the data and code are part of an ongoing study. Requests to access the datasets should be directed to e.peri@tue.nl.

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
