# Peer review of "Estimation of the Number of Active Sweat Glands Through Discrete Sweat Sensing"

_sensors, 2024, doi:10.3390/s24227187_

Round 1

Reviewer 1 Report

Comments and Suggestions for Authors

The paper presents a valuable contribution to the field of wearable sweat sensors and health monitoring. The authors effectively identified and addressed the research question of accurately estimating the number of active sweat glands using a discretized sweat-sensing device. The findings are significant and have potential implications for a wide range of applications, including health monitoring through sweat, and athletic performance analysis. Based on my review, I recommend the publication of this paper in the Sensors Journal after revisions to clarify some questions raised.

 1.     How were the parameters of the sweat gland model (e.g., sweat rate distribution, gland size, spatial arrangement) determined or estimated? Were these parameters based on experimental data or theoretical assumptions?

 2.     How were the model's assumptions validated or justified, especially regarding the representation of sweat gland heterogeneity and dynamic changes in sweat rate?

 3.     Regarding the sweat gland model, How were the model's assumptions validated or justified, especially regarding the representation of sweat gland heterogeneity and dynamic changes in sweat rate?

 4.     What factors were considered in the design of the discretized device model (e.g., sensor sensitivity, spatial resolution, noise characteristics)?

 5.     Were any simplifications made in the device model, and how might these affect the accuracy of the simulations?

 6.     How was the variability in sweat gland parameters (e.g., sweat rate, gland size, spatial arrangement) incorporated into the synthetic data?

 7.     What metrics were used to evaluate the algorithm's performance (e.g., accuracy, precision, recall, F1-score)? How were these metrics chosen, and why are they appropriate for the task?

 8.     Was a sensitivity analysis performed to assess the impact of noise, artifacts, and different device parameters on the algorithm's performance?

 9.     How did the algorithm's accuracy vary under different conditions? 

 10.  The conclusion part needs to be rewritten to bring more highlights to the paper.

Reviewer 2 Report

Comments and Suggestions for Authors

The paper introduces an interesting algorithm to estimate the number of active sweat glands and their individual sweat rates using discrete sweat sensing devices. This is an important step for improving the accuracy of biomarker analysis in sweat. The manuscript is well-structured, but there are areas that need clarification, further elaboration, and consideration of alternative approaches.

1.     The algorithm seems to face difficulties in high sweat rate scenarios. What are the specific conditions under which the error rate jumps above 50%, and how can these scenarios be minimized or handled better? Consider expanding this explanation for better understanding.

2.     The assumptions regarding synchronized sweat gland activation and uniform sweat rates across glands are critical. Could the authors elaborate on how realistic these assumptions are, and how deviations from these conditions might affect the performance of the algorithm? A sensitivity analysis would add depth to the discussion.

3.     The entire performance evaluation is based on synthetic data. What real-world experimental data or validation plans are in place? Mentioning any ongoing or future experimental validation would add credibility.

4.     The paper reports error rates, but it would be beneficial to present confidence intervals for these error rates to better understand the variability across different simulations. Additionally, does the algorithm consistently perform better for certain ranges of sweat rates or layouts?

5.     While the paper focuses on algorithmic and device development, are there any ethical considerations or practical challenges for implementing this technology in clinical settings? A brief discussion on safety, patient comfort, and data privacy would enhance the manuscript.
